

# Testing the effectiveness of different wash protocols to remove body surface contaminants in invertebrate food web studies

Melissa Jüds[1], Kerstin Heidemann[1], Bernhard Eitzinger[1,2] and Stefan Scheu[1]

[1] J. F. Blumenbach Institute of Zoology and Anthropology, Georg-August Universität Göttingen, Göttingen, Germany
[2] iES Landau, Institute for Environmental Sciences, RPTU Kaiserslautern-Landau, Landau in der Pfalz, Germany

## ABSTRACT

Molecular gut content analysis *via* diagnostic PCR or high-throughput sequencing (metabarcoding) of consumers allows unravelling of feeding interactions in a wide range of animals. This is of particular advantage for analyzing the diet of small invertebrates living in opaque habitats such as the soil. Due to their small body size, which complicates dissection, microarthropods are subjected to whole-body DNA extraction-step before their gut content is screened for DNA of their food. This poses the problem that body surface contaminants, such as fungal spores may be incorrectly identified as ingested food particles for fungivorous species. We investigated the effectiveness of ten methods for body surface decontamination in litter-dwelling oribatid mites using *Steganacarus magnus* as model species. Furthermore, we tested for potential adverse effects of the decontamination techniques on the molecular detection of ingested prey organisms. Prior to decontamination, oribatid mites were fed with an oversupply of nematodes (*Plectus* sp.) and postmortem contaminated with fungal spores (*Chaetomium globosum*). We used diagnostic PCR with primers specific for *C. globosum* and *Plectus* sp. to detect contaminants and prey, respectively. The results suggest that chlorine bleach (sodium hypochloride, NaClO, 5%) is most efficient in removing fungal surface contamination without significantly affecting the detection of prey DNA in the gut. Based on these results, we provide a standard protocol for efficient body surface decontamination allowing to trace the prey spectrum of microarthropods using molecular gut content analysis.

## INTRODUCTION

Ecosystems comprise a dense network of direct and indirect interactions between organisms and their biotic and abiotic environment, with feeding interactions forming the basis of food webs. Identifying those trophic interactions through direct observations in

Corresponding author
Melissa Jüds, mjueds@gwdg.de

the field are however difficult, especially when habitats are opaque, such as in soil, or includes animals that are very small, such as many invertebrate taxa. Under these circumstances feeding interactions can be identified using DNA-based analysis of regurgitates, faeces and the gut content (*King et al., 2008*; *Nielsen et al., 2018*; *Symondson, 2002*). As even low amounts of ingested food DNA can be detected through polymerase chain reaction (PCR) and subsequently be identified using species-specific primers or high-throughput sequencing, these molecular methods are both, highly sensitive and specific. Since their small body size complicates dissection, whole-body DNA extraction of small invertebrates, such as mites and springtails, is the usual method to obtain the gut content material. This poses the potential problem, that the DNA extract consists of DNA of various origin, *i.e.*, the DNA of the consumer, environmental DNA from the consumers body surface, the consumers symbiotic gut microbiome and DNA of ingested food. While the dissection of body parts, such as the abdomen or opisthosoma can significantly reduce excess DNA of the consumer (*e.g.*, *Krehenwinkel et al., 2016*) such an approach is inadequate to lower the amount of DNA on the body surface originating from spores, ectoparasites or faecal material. This mixture of DNA potentially leads to false assignments in the analysis of the consumers' diet. In addition, digestion processes and the short duration of the gut passage pose additional challenges for prey detection, but if overcome may allow unprecedented insight into trophic networks (*Agustí et al., 2003*; *Eitzinger et al., 2019*; *Read et al., 2006*).

Soils are a good example of an opaque habitat colonized predominantly by small invertebrates typically reaching high densities and diversity, such as springtails and mites. Most of these microarthropod species live as generalist feeders consuming a wide range of diets in particular fungi and bacteria (*Nielsen, 2019*; *Scheu & Setälä, 2002*). Microbial feeding species may even regulate the activity and thereby the functions and services of microorganisms, including litter decomposition, carbon sequestration and nutrient cycling (*Bardgett, 2005*; *Nielsen, 2019*). Molecular gut content analysis is a powerful method allowing to track trophic interactions of microbial feeders (*Gong et al., 2018*; *Jørgensen et al., 2005*). In order to understand the actual trophic relationships between microorganisms and microarthropods, we need to clearly distinguish between consumed microorganisms and non-food microbial material. This is particularly challenging since living and moving through the soil microarthropods may be covered with bacteria and fungi, in particular fungal spores (*Anslan, Bahram & Tedersoo, 2016*; *Renker et al., 2005*). Due to the high sensitivity of molecular gut content analysis, the detection of microorganisms attached to the body surface, may compromise the analysis of prey species in the gut of microarthropods and lead to wrong conclusions on the diet of soil-dwelling microarthropods.

Chlorine bleach (sodium hypochlorite, NaClO) is commonly used as a sterilizing agent in molecular laboratories and has been successfully applied to clean surfaces of insects of different life stages (*Davidson et al., 1994*; *Linville & Wells, 2002*) as well as bulk samples (*Greenstone et al., 2012*; *Hausmann et al., 2021*), and arachnids (*Miller-ter Kuile, Apigo & Young, 2021*) including oribatid mites (*Remén, Krüger & Cassel-Lundhagen, 2010*). Many of these studies also focused on the detectability of prey DNA in the gut (*Greenstone et al.*,

*2012*; *Linville & Wells, 2002*; *Miller-ter Kuile, Apigo & Young, 2021*; *Remén, Krüger & Cassel-Lundhagen, 2010*), but also on the detection of endosymbionts (*Meyer & Hoy, 2008*). Besides, these studies also aimed at working towards a standard method for surface decontamination of invertebrates for metabarcoding. However, the numbers of replicates and tested decontamination protocols were limited. In addition to the usage of chlorine bleach, various methods have been used for surface sterilization of invertebrates and plants, such as a combination of ethanol washing and UV irradiation in leafcutter ants (*Van Borm, Billen & Boomsma, 2002*), washing in a 5% formaldehyde solution of beetle eggs (*Duperchy & Zimmermann, 2003*), washing in acetone of mulberry aphid nymphs (*Fukatsu & Nikoh, 1998*) and washing in 0.01% Tween 20 and sonication for 5 min of plant leaves (*Burgdorf et al., 2014*). How efficient these sterilization treatments are on surface sterilization of soil microarthopods and how these affect the detection of gut contents remains to be tested. Therefore, we tested ten decontamination procedures and used 20 replicates in each of them with duplicated PCRs and triplicated PCRs when duplicates differed to evaluate technical and intraspecific variations. We chose the chemicals/methods based on literature focusing on using decontamination techniques on invertebrate body surfaces. Our experimental set up consisted of the oribatid mite *Steganacarus magnus* (Nicolet, 1855) as model microarthropod species, which was fed with bacterial feeding nematodes of the genus *Plectus* (*Plectus minimus* (Cobb, 1893) and *P. velox* (Bastian, 1865)) and was postmortem contaminated in the mycelium of the fungus *Chaetomium globosum* (Kunze, 1817). Both *S. magnus* and *Plectus* sp. are abundant in the litter and soil-layer of forests in Europe and *S. magnus* is known to feed on nematodes, especially *Plectus* sp. (*Heidemann et al., 2011*, *2014*). The study represents the first comprehensive test of surface decontamination methods for establishing a standardized procedure for molecular gut content analysis in soil microarthropods.

# MATERIALS AND METHODS

Portions of these texts were previously published as part of a thesis (*Jüds, 2023*).

## Sampling of animals

Soil- and litter-dwelling invertebrates were extracted in a laboratory over night from litter of a beech forest near the city of Göttingen (Lower Saxony, Germany) using heat extractors (*Kempson, Lloyd & Ghelardi, 1963*). A constant heating gradient of 40 °C above the samples to water cooled (15 °C) underneath the collecting vessels was applied. Living animals were collected in containers covered with wet tissue to prevent desiccation. Individuals of oribatid mite *Steganacarus magnus* (Nicolet, 1855) were identified using the identification key by *Weigmann (2006)* under a stereo microscope (Stemi 508; Zeiss, Jena, Germany) and starved for five days after which they were fed *ad libitum* with bacteria feeding nematodes (*Plectus minimus* (Cobb, 1893) and *P. velox* (Bastian, 1865), from nematode cultures) for three days. Starving and feeding took place under controlled temperature of 16 °C and constant darkness in a climate chamber. Thereafter, mites were checked under a stereomicroscope for attached nematodes to avoid false positives; no nematodes were attached. Mite individuals were stored separately at −80 °C in 1.5 mL

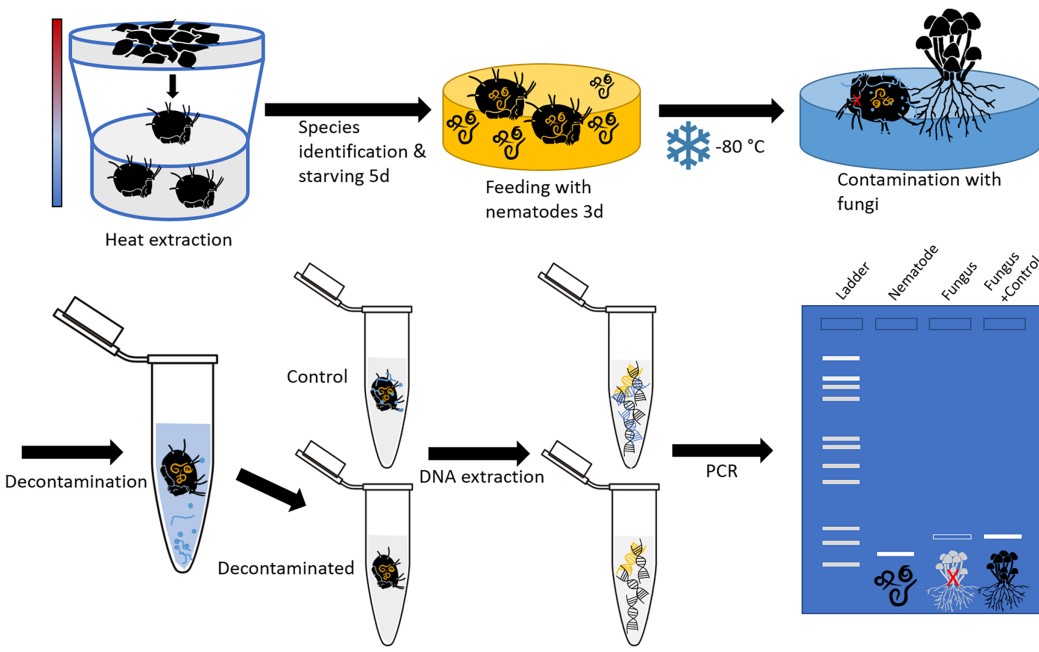

**Figure 1 Schematic representation of the experimental setup.** The oribatid mites used in this study were *Steganacarus magnus*. The nematodes fed to *S. magnus* were *Plectus* sp. and the fungus used for contamination of the mites was *Chaetomium globosum*. Primers amplifying the nematode genus *Plectus* and the fungus *C. globosum* were used for diagnostic PCR.

microcentrifuge tubes. Next, the body surface of dead oribatid mite specimens was then intentionally contaminated by rolling them over a colony of the fungus *Chaetomium globosum* (Kunze, 1817) cultivated on potato dextrose agar (PDA; Carl Roth, Karlsruhe, Germany). This was done by pushing the mites with a pair of forceps over a distance of at least 2 cm over the fungal colony, so that the animals' surface was visibly covered with fungal tissue. The contamination as well as the cleaning of animals was done under sterile conditions in a laminar flow hood. The experimental setup is shown in Fig. 1.

## Cleaning of the body surface

Cleaning of the body surface from fungal propagules (spores) was done by "washing" specimens using ten different methods (Table 1, Fig. 2). At least 20 individuals were tested with each method. For every washing step individual specimens were transferred into a new sterile 1.5 mL microcentrifuge tube and shortly vortexed after the next cleaning solution was added (500 µL). To transfer specimens to a new tube spring steel forceps were used, which were subsequently washed in absolute ethanol (EtOH; Carl Roth, Karlsruhe, Germany) and heat sterilized over a flame for several seconds to avoid cross-contamination. High grade RNase free water (Carl Roth, Karlsruhe, Germany) and absolute ethanol was used in the additional steps before and after the main decontamination procedure. A droplet of Tween 20 (Merck, Darmstadt, Germany) was added to water and decontamination liquids to break surface tension. After the last step of

**Table 1** The ten methods used for decontaminating the body surface of *Steganacarus magnus*.

| Method | Wash 1 | Decontamination | Wash 2 |
|---|---|---|---|
| Acetone + Tween 20 0.1% | X | Incubate in acetone; 5 min | X |
| Bleach 5% + Tween 20 0.1% | X | Incubate in bleach; 5 min | X |
| Flame | – | Pass through flame; 1 s | – |
| Formaldehyde 37% | X | Incubate in formaldehyde; 5 min | X |
| $H_2O_2$ (hydrogen peroxide) 30% | X | Incubate in $H_2O_2$; 5 min | X |
| Peracetic acid 2% | X | Incubate in peracetic acid; 5 min | X |
| SDS (sodium dodecyl sulfate) 0.1% | X | Incubate in SDS; 5 min | X |
| Sterillium® | X | Incubate in Sterillium®; 5 min | X |
| Ultrasound | X | Ultrasound bath; 10–30 s | X |
| UV Light | X | Treat with UV light; 30 min | X |

Note:
All washing steps (except flame treatment) were done in 1.5 mL sterile tubes and vortexed. 'Wash 1' included washing steps with $H_2O$ (2 min), followed by EtOH (absolute, 5 min) and $H_2O$ (2 min). 'Wash 2' included two $H_2O$ (2 min) washing steps. "–" indicates that this washing step was omitted.

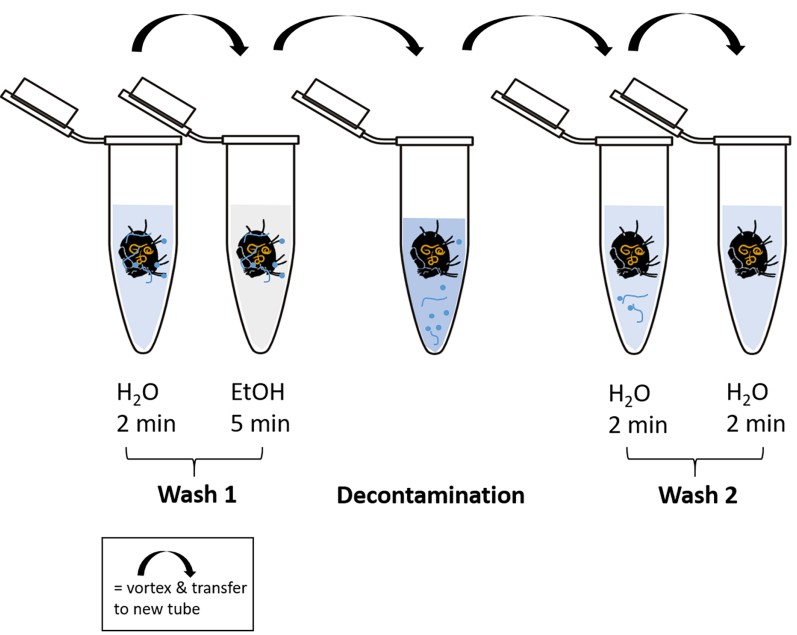

**Figure 2** Schematic representation of the decontamination process of fed and surface contaminated mites. RNase-free water and ethanol absolute were used for the washing steps. In each case, 500 µl of liquid was pipetted into a fresh microcentrifuge tube (1.5 mL) after transferring the individual with steel forceps. At each step, the tubes containing the liquid and the mite were vortexed for several seconds. The decontamination step includes the liquid according to the method used. After the last step of "Wash 2", the individuals were stored at −20 °C or directly processed for DNA extraction.

decontamination, each individual was transferred to a new sterile tube and either directly used for DNA extraction or stored at −20 °C until further usage. Information on molecular formula and manufacturer of decontamination reagents are listed in Table S1.

## DNA extraction

Whole body DNA extraction was done using the DNeasy Blood & Tissue Kit (Qiagen, Hilden, Germany) following the manufacturers protocol. Body tissue was mechanically crushed using steel pestles and tissue lysis was conducted on a heating and mixing platform (ThermoMixer®; Eppendorf SE, Hamburg, Germany) at 56 °C for 2 h. The final elution of DNA from the silica membrane was performed twice with 16 µL elution buffer "AE" (provided in the kit) per specimen, *i.e.*, DNA extracts were based on single specimens.

## DNA amplification

DNA amplification using PCR targeted three different gene regions. The first PCR targeted a 320 bp fragment of the D3 region (D3), part of the 28S rRNA gene, as a general invertebrate marker. This was carried out to verify the success of the DNA extraction and to exclude inhibition of the DNA amplification during PCR. Here, we used the primer pair D3A–5′-GACCCGTCTTGAAACACGGA-3′ and D3B–5′-TCGGAAGGAACCAGCTAC TA-3′ (*Litvaitis et al., 1994*; *Maraun et al., 2003*). For detecting contamination by *C. globosum* (Chae), the primer pair specific for *C. globosum* CHA F5–5′-GAGGTCACC AAACTCTTGATAATTT-3′ and CHA R6–5′-CCTACTACGCTCGGTGTGACAG-3′ targeting a 313 bp fragment was used. This primer pair was developed by using specific sequence data from own sequence data and data from the NCBI database (https://www. ncbi.nlm.nih.gov/) of the target fungal species and several non-target organisms. Primers were designed using the software Geneious (Version R10, https://www.geneious.com/) before conducting this experiment (B. Eitzinger, M. Jüds & K. Heidemann, 2017, unpublished data). To test if the decontamination procedure also affects the detectability of the ingested gut content, a primer pair targeting a 156 bp fragment of the 18S rDNA gene specific for the nematode genus *Plectus* sp. (Plec) was used: Plec-F-644– 5′-CTGRGATCC AAGGCTTATACTGC-3′ and Plec-R-799–5′-TAGARCCGTGGTCTTATTCT-3′ (*Heidemann et al., 2014*). PCR conditions included 34 cycles with denaturation of DNA double strands at 95 °C for 30 s, annealing of primer at 58 °C (D3), 59 °C (Chae), 62 °C (Plec) for 45 s, elongation of strands at 72 °C for 30 s. PCR started with heat activation of the Hot Start Taq polymerase at 95 °C for 15 min and ended with a final strand elongation at 72 °C for 10 min.

For all PCR reactions the SuperHotStar-Taq Master Mix was used (Genaxxon Bioscience GmbH, Ulm, Germany). PCR for the D3 fragment was carried out using 1 µL of each primer (100 pmol µL$^{-1}$; Eurofins Genomics, Ebersberg, Germany), 1 µL 25 mM MgCl$_2$, 1 µL of BSA (3%), 12.5 µL of 2× SuperHotStar-Taq Master Mix containing the polymerase and 2.5 µL DNA. PCR for *C. globosum* was carried out with 1 µL (100 pmol µL$^{-1}$; Eurofins Genomics, Ebersberg, Germany) of each primer and 12.5 µL of 2× SuperHotStar-Taq Master Mix (Genaxxon Bioscience GmbH, Ulm, Germany) containing the polymerase and 2.5 µL DNA. PCR for amplifying a 156 bp fragment of *Plectus* sp. was carried out using 2 µL (100 pmol µL$^{-1}$; Eurofins Genomics, Ebersberg, Germany) of each primer, 2 µL 25 mM MgCl$_2$, 2 µL of BSA (3%), 12.5 µL of HotStarTaq Master Mix Kit containing the polymerase and 2.5 µL DNA. In all reactions, RNase free water was used to fill up to 25 µL total reaction volume per sample.

PCR products were visualized and checked on a capillary electrophoresis system QIAxcel using AL320 as analyzing method (Qiagen, Hilden, Germany). This analyzing method is specialized on low PCR product concentrations and short fragment sizes (<500 bp). For each sample and primer pair two PCR reactions were carried out for the decontamination and gut content test. In case of inconsistent results of a sample, a third PCR was performed. A subset of purified PCR products were Sanger-sequenced (Microsynth Seqlab, Göttingen, Germany) for further validation of decontamination success or confirmation of positive detection of nematode and fungal DNA.

## Statistical analysis

Variations in the effectiveness of the decontamination procedures were inspected using analysis of variance (ANOVA). Differences between means were inspected using Tukey's honestly significant differences (HSD) test with the Holm correction method. For graphical presentation of the frequency of detection of *C. globosum* percentages were used illustrating the effectiveness of the different decontamination methods using the package ggplot2 in R (*Wickham, 2016*). All statistical analyses were performed using R (*R Core Team (2021)*, Version 4.2.1) and RStudio (RStudio 2022.07.0).

# RESULTS

## Decontamination effectiveness

All DNA extracts were tested positive for the D3 region and thus used for detection of fungal and nematode DNA.

Overall, the detection of the fungus *C. globosum* varied significantly with decontamination methods ($F_{10,436} = 19.45$, $P < 0.001$). In all non-decontaminated (=control) samples of *S. magnus* we detected DNA of *C. globosum* (Fig. 3). Generally, the detection of *C. globosum* varied significantly with decontamination methods ($F_{10,436} = 19.45$, $P < 0.001$). In eight of the ten decontamination methods the percentage of oribatids with detectable traces of *C. globosum* were significantly lower than the control: Acetone 67% *Chaetomium* sp. detection, hydrogen peroxide 55%, UV-light 50%, SDS 48%, peracetic acid 35%, ultrasound 35%, chlorine bleach 19% and formaldehyde 12%. In four decontamination methods, the detection of *C. globosum* dropped below75% compared to the control (formaldehyde, chlorine bleach, peracetic acid and ultrasound). Detection of *C. globosum* was lowest in bleach and formaldehyde treated individuals with a detection rate of 19% and 12%, respectively.

## Decontamination and detection of gut content

Detection of nematode prey *Plectus* sp. in the gut of *S. magnus* varied significantly with decontamination methods ($F_{10,436} = 7.98$, $P < 0.001$). Maximum detection success of *Plectus* sp. was over 50% with highest detection rates in non-decontaminated (53%) and flame treated *S. magnus* (55%; Fig. 4). Overall, detection frequency of *Plectus* sp. was not significantly reduced after decontamination with flame (55%), bleach (49%), SDS (48%) and formaldehyde (36%). The other decontamination methods (hydrogen peroxide,

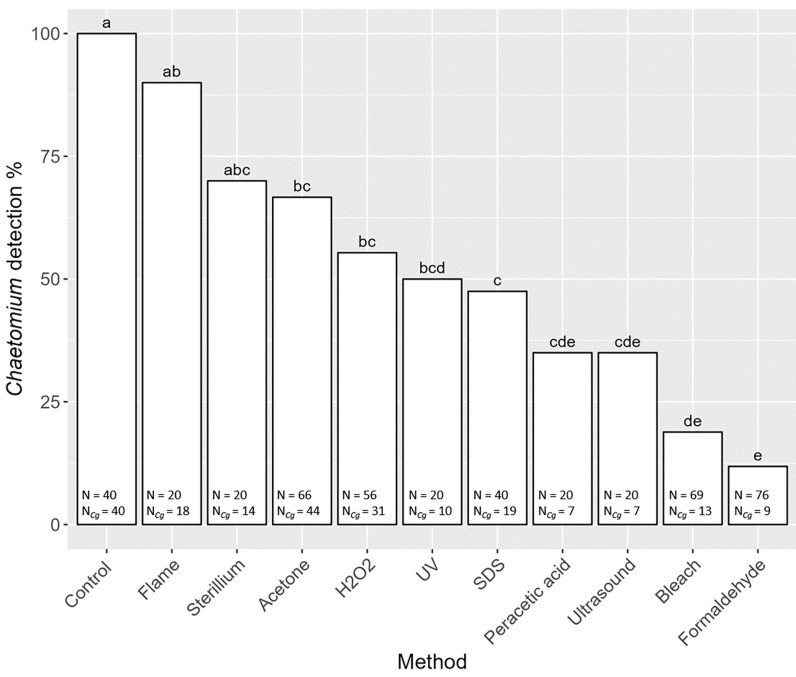

**Figure 3 Effects of the method of decontamination on the detection frequency (%) of the fungus *Chaetomium globosum* in *Steganacarus magnus*.** Bars sharing the same letter do not differ significantly ($p < 0.05$; Tukey's HSD test). Total numbers of tested individuals (N) and total numbers of detected *C. globosum* ($N_{Cg}$) are given at the bottom of the respective bars. For details of the decontamination methods see Table 1 and Methods.      

Sterillium, ultrasound and acetone) reduced the detection of *Plectus* sp. to below 20%. In two decontamination treatments (peracetic acid and UV) *Plectus* sp. was not detected.

## DISCUSSION

We tested ten methods for their surface decontamination effectiveness of a soil dwelling oribatid mite species, *S. magnus*, in a controlled laboratory experiment. Treatments with 5% bleach and 37% formaldehyde reduced contamination with fungal DNA significantly. Even though six other methods also reduced the detection of fungal DNA in contaminated *S. magnus* individuals, the reduction rate varied and was overall significantly less effective. Additionally, we tested if the decontamination methods affect the detection of prey DNA in the gut of *S. magnus via* species-specific PCR. Our results indicate that the detection of nematode DNA (*Plectus* sp.) is affected by the type of decontamination method. The detection of prey DNA was not significantly affected in specimens treated with bleach, whereas PCR amplification was reduced in specimens treated with formaldehyde. This suggests that even short exposure to formaldehyde interferes with DNA in the gut content, lowering the chance of prey DNA detection. Therefore, the treatment with bleach represents the best decontamination procedure for soil arthropods to be used for molecular gut content analysis. Other methods that did not affect the detection of gut DNA, such as flame and SDS, did not remove surface contamination or did not reduce sufficiently fungal contamination.
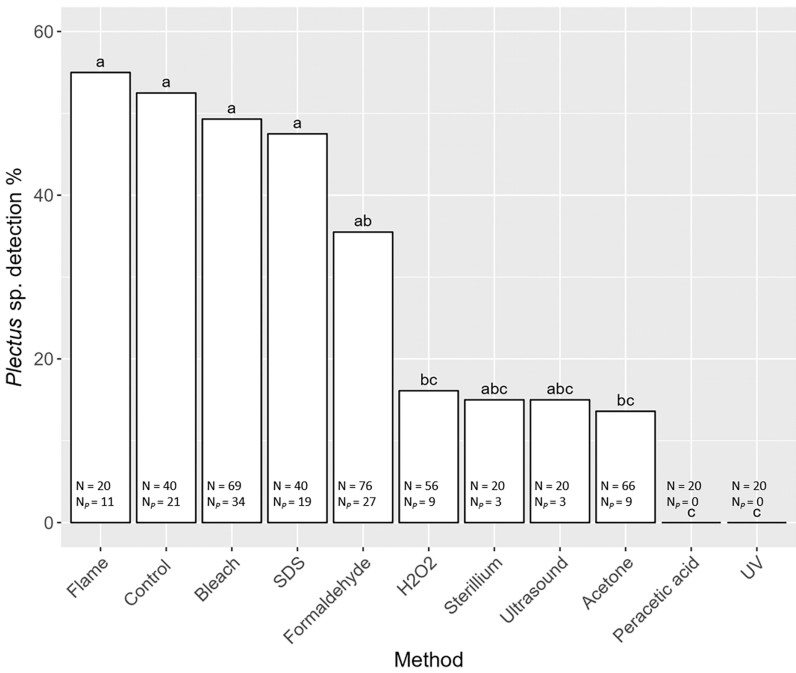

**Figure 4 Effects of the method of decontamination on the detection frequency (%) of *Plectus* sp. in the gut of *Steganacarus magnus*.** Bars sharing the same letter do not differ significantly ($p < 0.05$; Tukey's HSD test). Total numbers of tested individuals (N) and detected *Plectus* sp. ($N_P$) are given at the bottom of the respective bars. For details of the decontamination methods see Table 1 and Methods.

Our experimental set up with *S. magnus* feeding on *Plectus* sp. as nematode prey has been used previously in laboratory feeding experiments as well as in field studies investigating the gut content of soil microarthropods (*Heidemann et al., 2014*).
As *S. magnus* can be collected in high numbers this allowed for a solid replication of decontamination methods and to establish a reliable experimental basis. Mites of the genus *Steganacarus* are characterized by a strongly sclerotized globular and ptychoid body shape and thus are simple to handle and do not break easily during handling. Additionally, the notogaster of *S. magnus* is covered with several long setae on which microbial propagules are likely to stick. When handling soil animals from other taxa such as Collembola and Nematoda, which possess a soft body cuticle and/or have a smaller body size, the protocol for decontamination may need further refinement, such as the use of low bleach concentrations or shorter incubation times. The experimental set up and results of the present study may be taken as starting point to evaluate a wider range of body surface contaminants as well as primers targeting to amplify the gut content of decomposer soil arthropods.

For analyzing the fungal diet of oribatid mites *Remén, Krüger & Cassel-Lundhagen (2010)* used a decontamination protocol containing 3.7% bleach, but only obtained sufficient ingested fungal DNA material after pooling dissected guts. The dissection of guts is time-consuming and challenging for screenings of large numbers of soil microarthropod species for their gut content, since time between sampling and freezing of organisms is an

important factor in the quality of gut DNA extracts (*Alberdi et al., 2019*). Additionally, as these authors also stated, dissecting the gut is not feasible for small or delicate microarthropods. Therefore, we tested the decontamination methods in combination with whole body DNA extraction and diagnostic PCR. *Greenstone et al. (2011)* suggested to separate individuals from a collection vessel with many animals using a brush to prevent cross-contamination. While this might be a suitable method for handling large invertebrates living above the ground, it is not an option for tiny animals living in opaque habitats such as the soil. Thus, *Greenstone et al. (2012)* also tested different bleach concentrations for their decontamination success and found that 2.5% bleach and 40 min rotated incubation provide the best results. In our testing trial we found that 5 min of treatment with 5% chlorine bleach is sufficient, additionally allowing for an overall faster DNA extraction process which potentially benefits the recovery of prey DNA in the gut content. However, since their study focused on insects living aboveground, we anticipated the procedure to be too harsh for small soil animals. Similarly, *Miller-ter Kuile, Apigo & Young (2021)* suggested context specific decontamination treatments for DNA metabarcoding of invertebrates. The authors stressed the need for a standardized and well tested method for surface decontamination of invertebrates. We shared this aim and tested a wide range of methods in a comprehensive and replicated manner aiming at establishing such standardized method.

Considering the enormous diversity of arthropod morphology and physiology, establishing standard procedures for surface decontamination is challenging. This applies especially for studies aiming at analyzing the gut content of microbe- and fungivorous animals in metabarcoding studies. In those, a decontamination of the body surface of consumer species before DNA-extraction may be crucial to limit the amplification and consequently the read number of non-target species. In many metabarcoding studies, singletons and sequences which make up only a low number are thought to originate from contamination. Consequently, these are excluded in the bioinformatic pathway in order to rule out incorrect results by including false positives (*Cuff et al., 2021*). Using our protocol, metabarcoding studies will be able to eliminate surface contaminants but keeping sequences from rare feeding interactions, thus allowing for a more in-depth analysis of food web interactions in soil systems.

Our results on the decontamination effectiveness of ten methods suggest that the best practice for a reliable surface sterilization with minimal effects on the detection success of food DNA in the gut content in mites is a treatment with bleach. The treatment with formaldehyde (37%) had sufficient decontamination results but might harm the gut content DNA. The use of other decontamination reagents, such as SDS and acetone, or treatments with UV radiation, did not result in valid decontamination of the body surface. We advise a standardized protocol consisting of three essential steps: "Wash 1"-rinsing and washing with water and ethanol to rinse off easy to remove contaminants, "Decontamination"-incubation in bleach (5%) for 5 min, "Wash 2"-rinsing with water to remove residual bleach and contaminants. The concentration of bleach may be reduced to 1.5% for delicate microarthropods such as Collembola and additionally the incubation time can be shortened as well. Only sterile tubes and tools should be used in a clean

working environment, *e.g.*, under a laminar flow hood. How animals are transferred or how vigorously animals are mixed in the solution (by vortexing or gently inverted by hand) should be considered depending on the animals' surface structure and morphology. While further refinement and adjustment may be necessary for decontamination of other more delicate microarthropods, our method here provides a standardized protocol for the decontamination of a wide range of microarthropod taxa.

## ACKNOWLEDGEMENTS

We thank Kai Butzelar for conducting laboratory experiments and molecular work. We also thank Svenja Meyer for permission to use the schematic icons of *Steganacarus magnus*, nematodes and fungi.

### Funding

This work was funded by the German Research Foundation (DFG Priority Program 1374 SCHE 376/38-1). The funders had no role in study design, data collection and analysis, decision to publish, or preparation of the manuscript.

### Grant Disclosures

The following grant information was disclosed by the authors:
German Research Foundation: 1374 SCHE 376/38-1.

### Competing Interests

The authors declare that they have no competing interests.

### Author Contributions

- Melissa Jüds conceived and designed the experiments, performed the experiments, analyzed the data, prepared figures and/or tables, authored or reviewed drafts of the article, and approved the final draft.
- Kerstin Heidemann conceived and designed the experiments, performed the experiments, authored or reviewed drafts of the article, and approved the final draft.
- Bernhard Eitzinger conceived and designed the experiments, authored or reviewed drafts of the article, and approved the final draft.
- Stefan Scheu conceived and designed the experiments, authored or reviewed drafts of the article, and approved the final draft.

### DNA Deposition

The following information was supplied regarding the deposition of DNA sequences:
The short PCR products for the fungal contaminant *Chaetomium globosum* and for the diet/prey *Plectus* sp. are available at GenBank: OQ955516 to OQ955536 (*C. globosum*) and OQ955538 to OQ955545 (*Plectus* sp.).
https://www.ncbi.nlm.nih.gov/nuccore/?term=OQ955516 to OQ955536.
https://www.ncbi.nlm.nih.gov/nuccore/?term=OQ955538 to OQ955545.

## Data Availability

The short PCR products for the fungal contaminant *Chaetomium globosum* and for the diet/prey *Plectus* sp. are available at GenBank: OQ955516 to OQ955536 (*C. globosum*) and OQ955538 to OQ955545 (*Plectus* sp.).

The raw sequences are available in the Supplemental File.

## Supplemental Information

Supplemental information for this article can be found online at http://dx.doi.org/10.7717/peerj.16018#supplemental-information.

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
