# Peer review of "Testing the effectiveness of different wash protocols to remove body surface contaminants in invertebrate food web studies"

_PeerJ, doi:10.7717/peerj.16018_

## Round 0.1 · original submission · Major Revisions

There are minor but numerous amendments of both reviewers and that is why my decision is "major revisions".

·

Basic reporting

The article provides information that is very necessary in this type of research.
The article is well written under the parameters established by the journal.
The figures are clear and graphically support the manuscript.
In the tables, it is necessary to review Label of Table 1, the name of the species must be written in italics (Steganacarus magnus), as they have done in other parts of the text.
Check line 96, there is a repeated word "in"
Lines 102 and 104 review, as the first mention of the species include the name of the author.

Experimental design

No comment

Validity of the findings

No comment

Additional comments

The article provides information that is very necessary in this type of research.
I particularly like the figures they are clear and fun to look at and graphically support the manuscript.

·

Basic reporting

This study tackles an important question within dietary analysis and trophic ecology. Other studies (highlighted by this one) have done so on smaller scales or less systematically in the past. The methods and results are straightforward and elegant, lending to an accessible narrative which will surely be a convenient reference point for many others using these methods. Thanks for your hard work in putting this together!

The context within which this specific work was carried out (i.e., soil microarthropods) is particularly compelling and more could possibly be made of this in the title/abstract (if only the title referring to ‘invertebrate food web studies’ given that this is unlikely to be relevant to any vertebrate consumers). There are just a few other minor suggestions for clarity below.

Experimental design

The experimental design is elegant, straightforward and nicely addresses the aims of the study. It might be worth adding some information about how the different concentrations of decontamination solutions were decided (maybe to Table 1). Were these based on the literature, or decided by other means? It would be interesting to see how concentration and time affect the outcomes, but that would appreciably be a massive undertaking for one study and the relationships between figures 3 and 4 indicate that the optimal methods would likely remain the same.

Is the fungus on which the nematodes were reared known and is there any chance of it being the same species as was used for the intentional contamination? If this is known, it would be a nice way of ruling out secondary detection of fungus in the gut rather than the exterior of the mites.

Lines 159-160: It would be good if the process of designing these primers was briefly described (e.g., whether it was based on mass alignments or specific sequence data; whether any particular software was used).

Lines 112-113: It would be nice to have a very brief description of how the heat extractors work in the main text. My interpretation is that this is similar to a Tullgren funnel. Was the heat extraction performed in the field? What were the mites collected into?

Lines 180-181: It isn’t immediately clear from Qiagen documentation what AL320 refers to – it might be valuable to briefly state the cartridge used and any other key details.

Lines 189-191: Were these visualized using ggplot2? If so, it would be useful to cite the package.

Validity of the findings

The findings are clearly reported and nicely presented. I do think there is a nice opportunity to discuss this briefly in the context of the wider literature around false positives and false negatives and the epistemology of metabarcoding. False positives are a significant problem in molecular dietary analyses (which is stated in the manuscript), but we typically deal with these in bioinformatics by removing low abundance sequences. This, however, removes a lot of legitimate detections, resulting in false negatives. The balance between these is incredibly fine and it’s almost impossible to avoid both. This study shows this same problem, but pre-sequencing. There are likely instances in which false negatives or false positives are more important. For example, if looking at the diet of a whole population, conservative data with many false negatives could be better since these will likely balance across the population, whereas for a small number of individuals, it may be inverted, although even this would likely be debated by many. As well, for metabarcoding data if read counts are used, false positives will likely be less problematic given that they are often less abundant, whereas presence-absence data will represent them more absolutely. This doesn’t need much discussion, but I think it’s very relevant here in choosing an optimal decontamination method and interpreting the outcomes. In a more direct link, the decontamination may also reduce the prevalence of false positives in metabarcoding data, reducing the burden on bioinformatics to remove these detections.

For a more direct comparison of the results in figures 3 and 4, it could be worth considering a plot of the ratio of Plectus to Chaetomium detections (so, the optimal solutions would be the highest values). This may be superfluous, but I thought I would suggest it for consideration at least.

Line 198: For ease-of-reading, it could be valuable to add ‘the fungus’ before C. globosum to remind the reader.

Lines 222-224: I’m not sure what is meant by ‘statistically less effective’ here. Would ‘significantly less effective’ be accurate?

Line 244: Is there any advice/guidance you can offer for refining the process for other taxa? This is eluded to in the summary on lines 282-284, which could be repeated here.

Additional comments

Line 1: Is ‘efficiency’ the correct word here? I think effectiveness or efficacy may be more fitting. This is true throughout (e.g., line 37).

Lines 33-34: In some cases, just part of the body can be used (e.g., abdomen).

Lines 35-36: It might be worth specifying ‘for mycophagous species’ after fungal spores, since many common predator-prey or herbivore-plant interactions are not concerned with the detection of fungal DNA.

Lines 59-61: In some cases, especially with specialised equipment, gut dissection is possible, or sections of the body (e.g., abdomen) could be used, which might be worth representing here.

Line 100: “Up to three PCR replicates” sounds quite ambiguous. I think it may be clearer to say “PCRs were duplicated, and triplicated when duplicates differed.” or something to that effect.

---

## Round 0.2 · accepted · Accept

The authors considered all comments of both reviewers and that is why this version can be accepted for publication.